# The Impact of Transcription Factor Prospero Homeobox 1 on the Regulation of Thyroid Cancer Malignancy

**DOI:** 10.3390/ijms21093220

**Published:** 2020-05-02

**Authors:** Magdalena Rudzińska, Barbara Czarnocka

**Affiliations:** 1Department of Biochemistry and Molecular Biology, Centre of Postgraduate Medical Education, 01-813 Warsaw, Poland; magdda.rudzinska@gmail.com; 2Institute of Molecular Medicine, Sechenov First Moscow State Medical University, 119991 Moscow, Russia

**Keywords:** PROX1, lymphatic factors, thyroid cancers

## Abstract

Transcription factor Prospero homeobox 1 (PROX1) is continuously expressed in the lymphatic endothelial cells, playing an essential role in their differentiation. Many reports have shown that PROX1 is implicated in cancer development and acts as an oncoprotein or suppressor in a tissue-dependent manner. Additionally, the PROX1 expression in many types of tumors has prognostic significance and is associated with patient outcomes. In our previous experimental studies, we showed that PROX1 is present in the thyroid cancer (THC) cells of different origins and has a high impact on follicular thyroid cancer (FTC) phenotypes, regulating migration, invasion, focal adhesion, cytoskeleton reorganization, and angiogenesis. Herein, we discuss the PROX1 transcript and protein structures, the expression pattern of PROX1 in THC specimens, and its epigenetic regulation. Next, we emphasize the biological processes and genes regulated by PROX1 in CGTH-W-1 cells, derived from squamous cell carcinoma of the thyroid gland. Finally, we discuss the interaction of PROX1 with other lymphatic factors. In our review, we aimed to highlight the importance of vascular molecules in cancer development and provide an update on the functionality of PROX1 in THC biology regulation.

## 1. Introduction

Transcription factor Prospero homeobox 1 (PROX1) is a homolog of the Prospero in Drosophila that regulates the development of various organs, including the central nervous system [1], lens, retina, [2], liver [3], heart [4], pancreas [5], and cell fate of lymphatic endothelial cells (LECs) [6]. The *PROX1* gene encodes a protein belonging to the Homeobox family, which has a characteristic Prospero domain at the C-terminus [7].

It has recently been established that PROX1 has a variety of roles in cancerogenesis, and its functions may change according to the type of tissue. Thus, PROX1 acts as a tumor suppressor in hepatocellular carcinoma [8], esophageal cancer [9], pancreatic cancer [10], oral cancer [11], hematologic malignancy [12], sporadic breast cancer [13], carcinoma of the biliary system [14], and papillary thyroid cancer (PTC) [15]. On the other hand, PROX1 promotes aggressive behavior of colorectal cancer [16], kaposiform hemangioendothelioma [17], glioma [18,19], and our last observations point to its distinct oncogenic role in follicular thyroid cancer (FTC) [20,21,22].

Thyroid cancer (THC) is the most common malignancy of the endocrine system, which is clinically divided into categories: (1) well-differentiated thyroid cancer (DTC), including PTC and FTC carcinomas, (2) poorly differentiated thyroid cancer (PDTC) (3) undifferentiated–anaplastic thyroid cancer (ATC), and (4) neuroendocrine C-cell derived-medullary thyroid cancer (MTC) [23]. Squamous cell carcinoma of the thyroid gland (SCT) is an unusual neoplasm, which is thought to arise as a primary tumor or as a component of anaplastic or undifferentiated carcinoma [24]. 

Depending on the histological variant, THC can use the different vascular routes to metastasize. Here, PTC spreads preferentially to the lymph nodes via the lymphatic system, and the more aggressive types, FTC, MTC, ATC, and SCT, tend to metastasize to distant organs (such as lung and bone) through the bloodstream [25,26].

Recent data strongly indicates that the cancer cell road of metastasis is highly connected with active vascular factors in the microenvironment and their expression in the cancer cells [27]. In this scenario, many molecules associated with blood endothelial cells (such as CD44, Intercellular adhesion molecule 1; ICAM1, Vascular endothelial growth factor receptor 1; VEGFR-1 and Neutropilin-1) and LEC-specific proteins (including Podoplanin; PDPN, Lymphatic vessel endothelial hyaluronan receptor 1; LYVE-1, VEGFR-3, VEGFC, and PROX1) can be expressed in tumor cells and peri-/intra-tumoral vessels consequently regulating angiogenic potential of tumors. Furthermore, the altered expression and secretion of vascular molecules in the tumoral surrounding can change the behavior of cancer and stromal cells; as a result, increasing the invasiveness and metastasis of tumors [27]. 

Interestingly, VEGFR-1 is expressed in blood vessels of the tumor but not in those of the healthy tissue [28]. VEGFD can induce both intra- and peri-tumoral lymphatic vessel development; however, it is not involved in lymph node metastasis [29]. Finally, a series of lymphatic factors, including PROX1, VEGFC, PDPN, VEGFR-3, SOX18 (SRY-Box transcription factor 18), and COUP-TFII (Nuclear receptor subfamily 2 group F member 2) can be expressed in tumoral cells and, consequently, control their properties, such as invasion, migration, proliferation, survival, epithelial to mesenchymal transition (EMT), and adhesion [30,31,32,33,34,35].

In the presented review, we discuss the role of PROX1 in THC development and recent advances in this field. We describe the PROX1 mRNA / protein sequences, PROX1 expression pattern in THC and its epigenetic regulation. Next, we present the most important biological processes and genes regulated by PROX1 in CGTH-W-1 (squamous cell carcinoma of the thyroid derived; SCT) based on internal RNA sequencing analysis. Finally, the last segment of the review details the transcriptional interaction of PROX1 with other lymphatic markers, such as VEGFC, VEGFR-3, and PDPN.

## 2. Thyroid Cancer Classification

Most primary THCs are epithelial tumors that originate from thyroid follicular cells and can appear as three main histopathological types of carcinoma: PTC, FTC, and ATC. PTC accounts for 85%–90% of all thyroid cancer cases, followed by FTC; 5%–10% [23]. ATC accounts for less than 2% of thyroid cancers, and it is a lethal malignancy (survival ~6 months from diagnosis), typically arising in elderly patients [36]. Next, MTC, with estimated prevalence maximally 2% of THC cases, is a form of thyroid carcinoma originating from thyroid parafollicular (C) cells with the characteristic presence of numerous endocrine secretory granules containing calcitonin in the cytoplasm [37]. SCT is extremely rare (<1%) and carries the unfortunate prognosis of thyroid malignancy, which often mixes with heterogeneous elements and is associated with areas of well-differentiated PTC or FTC [24].

The recommended treatment for low-risk PTC patients (females < 45 years old with tumor limited to the thyroid gland) is thyroid lobectomy (removal one of two thyroid lobes) followed by thyroid stimulating hormone (TSH) suppressive therapy [38]. High-risk patients (males and women > 45 years of age with high-grade tumors) are subjected to total thyroidectomy (removal of the entire thyroid gland) followed by radioactive iodine (131I) ablation [39]. PTC spreads relatively easily to the neck, but distant metastases are found only in 1% of patients, mostly in lung and bones [40]. The 5-year survival rate for PTC patients accounts for 100% of patients with localized tumor, 99% for regionally spread cancer, and 78% for distant metastasizing, which is better than for FTC cases, where the rate is 100% for the localized tumor, 96% for regionally spread, and 63% for distant metastases. FTCs spread to remote sites, primarily to bones, and the most common treatment of FTC is total thyroidectomy [41].

In most aggressive types, the total thyroidectomy gives the best chance of cure for patients with MTC [42], and for ATC, the treatment is usually palliative with radiotherapy [43]. The 5-year survival rate for metastatic MTC and ATC is 39% and 4%, respectively [41].

In the context of genetic alterations, the most frequent and mutually exclusive genetic changes in PTCs are BRAF V600E, RAS, and RET/PTC rearrangement, leading to constitutive activation of the signaling pathway of mitogen-activated kinases (MAPK) (Figure 1) [44]. 

*BRAF* occurs in 45% of PTCs, and in the majority, it is a substitution at the second position of codon 600 (V600E; GTG > GAG), c.1799 T > A) resulting in an amino acid change from valine to glutamic acid that leading to constitutive activation of serine/threonine kinase BRAF [46]. 

The *RAS* gene encodes a family of three highly homologous oncogenes: NRAS, HRAS, and KRAS, in which mutations occur in 10%–20% of PTCs [47]. RAS proteins transmit the signals from the receptors on cell membranes to several types of targets in the cell controlling MAPK and 3-phosphatidylinositol kinase PI kinase (PI3K) signaling pathways [47]. All point mutations of the RAS gene fix the protein activated states and, therefore, result in continuous stimulation of downstream targets of RAS [48].

The RET/PTC rearrangements occur in the chimeric oncogene RET/PTC, where the C-terminal kinase domain of the RET transmembrane tyrosine kinase receptor is fused to one of the different upstream partners, resulting in constitutive RET activity. At least 12 rearranged forms of the RET gene have been isolated and detected in 30% of the PTC cases, from which RET/PTC1 and RET/PTC3 are the most common [44,49].

In contrast, the characteristic genetic modifications that are mutually exclusive in FTC are changes in the RAS, PTEN, and PIK3CA genes, as well as rearrangement of PAX8-PPARγ, activating the 3-phosphatidylinositol kinase PI kinase (PI3K/Protein Kinase B–AKT) [44,50]. 

The RAS mutations were observed in approximately 40%–50% of FTC cases [51], and the modification was predominantly found in the NRAS codon 61, which positively associated with distant metastases of FTC [52].

Following, the mutation or deletion of the tumor suppressor gene – PTEN (phosphatase and tensin homolog) and PIK3CA transcript (coding the p110α catalytic subunit of PI3K) are the classical genetic alterations that activate the PI3K–AKT pathway in ~10% and 10%–30% of FTC cases, respectively [53,54]. 

The PAX8 gene encodes a transcription factor required for the generation of thyroid follicular cells and tissue-specific gene expression in the thyroid gland. The PAX8/PPAR fusion results in significant increases in expression of PAX8/PPAR chimeric protein and, as a result, inhibits the tumor suppressor activity of PPAR [55]. The PAX8/PPAR rearrangement presence was detected by real-time PCR in ~35% of FTC cases [56].

Consequently, the constitutive activation of pathways associated with the molecular changes in PTC and FTC can give rise to the formation of more aggressive forms PDTC and ATC [45]. The sporadic MTC development is mainly connected with RET receptor tyrosine kinase mutation, which has an essential role in cell survival, differentiation, and proliferation [37]. However, in ~10%–30% of patients, the RAS mutation was also found [57,58].

Additionally, through whole-genome sequencing, in many malignant THC cases, the mutations in the promoter region of telomerase reverse transcriptase (*TERT*) were found, contrary to the early stages of thyroid tumors [59]. *TERT* transcript is a 35 kb gene located on chromosome 5, which contains 16 exons and a 330 base pair promoter region. Two main *TERT* mutations: 1 295 228 C>T (C228T) and 1 295 250 C>T(C250T) can increase the TERT transcriptional activities. Particularly prevalent in THC is the C228T variant, which appeared in Liu X. et al.’s, 2013, analysis with 11.7% of PTCs, 11.4% of FTCs, 37.5% of PDTCs, and 42.6% of ATCs [59]. Moreover, the coexistence of *TERT* with *BRAF* or *RAS* alterations had a synergistic effect on poor clinicopathologic outcomes of PTCs, such as disease recurrence and patient mortality [60]. All data suggest that *TERT* promoter mutations may play a role in the THC de-differentiation, progression, and aggressive behavior [61]. Interestingly, Lee et al., 2019 correlated *TERT* and *PROX1* mRNA expression levels in several cancer types, including melanoma, esophageal and head and neck, and lung cancer, with PROX1 downregulation indicating a poorer prognosis in melanoma [62]. The authors concluded that PROX1 perhaps regulates TERT in an activity-dependent manner with other genetic changes. The PROX1:TERT relation can be a new scientific aim, which has to be elucidated in THC research.

The establishment of molecular changes in THC and recent progress in this field provide unprecedented opportunities for the development of molecular-based diagnostic, prognostic, and therapeutic strategies for a different type of THC.

## 3. PROX1 mRNA/Protein Isoforms and Antisense of PROX1 Characterization

PROX1 is a transcription factor essential for the embryogenesis of a variety of organs. The human PROX1 transcript is located on chromosome 1q32.2–q32.3 composed of five exons and four introns and produces two variants: NM_002763 and NM_001270616, both encoding the same protein product, but the NM_002763 transcript is longer by 322 nucleotides [7]. Long noncoding transcript (*PROX1-AS1*) transcribed from the antisense strand of PROX1 is located on chromosome 1q32.3 with transcript length 3399 bp (Figure 2A). 

The existence of different mRNA isoforms (7.9 kb and 2.9 kb) coding various PROX1 protein forms were presented by Zinovieva et al., 1996 and Dudas et al., 2008 using sequencing and hybridization [7,63]. In the performed research, the dominance of the 7.9 bp form was correlated with hepatocellular carcinoma samples, while the shorter isoform 2.9 kb was exclusively detected in cholangiocellular carcinoma [63]. 

The PROX1 protein contains 737 amino acids with a molecular weight of 82.3 kDa. Structurally, the PROX1 protein includes a unique homeodomain followed by a conserved prospero domain at the C-terminus and two nuclear receptor boxes (NR boxes) with nuclear localization signal (NLS) at the N-terminus (Figure 2B). PROX1 can act as a transcriptional activator, transcriptional repressor, or a transcriptional corepressor. While Prospero-/homeodomain and NLS are responsible for DNA binding, the NR boxes can interact with nuclear receptors, e.g., HNF4a/NR1A1 or SF-1/NR5A1 [64].

## 4. PROX1 Expression in Thyroid Cancer

Choi D. et al., 2016 described that PTC specimens show a consistent downregulation of PROX1 by more than 2-fold (*p* < 1 × 10^−4^) compared to normal thyroid tissues, which was confirmed by our simulation (with GEPIA database) using another set of thyroid cancer gene profiling studies (Figure 3).

Authors suggested that PROX1 downregulation can be already detectable in follicular and oncocytic adenomas, implying that this genetic event may happen in the early stage of follicular carcinogenesis [15]. 

Next, PROX1 downregulation in PTC-derived cells (BcPAP, TPC-1), normal thyroid cells (Nthy ori-3-1), ATC-derived cells (8505C), and two FTC-derived cell lines (FTC236 and FTC238) in comparison to FTC-derived cell line (FTC-133), and SCT cells (CGTH-W-1) was observed [22]. Moreover, as we found, lower PROX1 expression levels correlated with more prolonged survival and reduced disease severity, i.e., with I and II grades in comparison to III and IV stages. The switch of the PROX1 expression level in different cancer stages was suggested to be negatively regulated by fibroblast growth factor 2 (FGF2) [22]. In lens epithelial cells (LCs), the positive PROX1–FGFR signaling feedback loop was demonstrated, leading to PROX1 upregulation in response to FGF2 [65]. Additionally, the positive stimulation of PROX1 by FGF2 in CGTH-W-1 cells was noticed [22]. Interestingly, the lower expression of *PROX1* mRNA in THC tissues and cultured cancer cells did not correspond to the protein level that revealed accumulation in the cytoplasm [15,20]. This phenomenon was connected with the higher stability of the cytoplasmic form of PROX1 protein. The experiments performed with Kaposi sarcoma cells showed that PROX1 mRNA contains a canonical AU-rich element (ARE) in 3’-untranslated region (3′-UTR) facilitating binding of RNA binding protein HuR which stabilizes the transcript [66].

## 5. Epigenetic Regulation of PROX1

Genetic and epigenetic mechanisms can be involved in PROX1 expression regulation [11,13]. Mutations [67], DNA methylation [11], and non-coding RNA [68] appear to be the major mechanisms modulating the PROX1 function.

### 5.1. Mutations and Methylation

Post-transcriptional RNA editing is a process in which the nucleotide sequence of a nuclear mRNA is changed from that encoded in genomic DNA. RNA editing occurs through base modification, by deamination of cytidine (C) to uridine (U) or by deamination of adenosine (A) to inosine (I), in nuclear mRNA. Uridine and inosine are recognized by translational apparatus as thymidine and guanosine, respectively, so the net effects are changes in C-to-T and A-to-G [69]. In this context, the A-to-G mutation affects PROX1 function in human specimens of pancreatic, colon, and esophageal cancers and A to I in esophageal cancer. All detected variations were observed in cDNA PROX1 but not at the genomic DNA level [67,70]. Still, no experimental data provide information on *PROX1* mutations in THC and their effect on PROX1 function. Importantly, several single nucleotide polymorphisms (SNPs) present in intronic regions of *PROX1* were suggested to modulate *PROX1* expression levels with potential involvement in the pathogenesis of type 2 diabetes [71]. Therefore, it cannot be excluded that modulation of *PROX1* expression levels by these SNPs will also influence THC pathogenesis and/-or aggressiveness.

DNA methylation is one of the most common epigenetic modifications in mammals, and in healthy cells, it ensures the proper regulation and stable gene silencing. DNA methylation that is catalyzed by DNA methyltransferases (DNMTs) is associated with the addition of a methyl group to cytosine residue present within CpG dinucleotides, which are concentrated in large clusters (CpG islands) [72]. It is commonly known that inactivation of specific tumor suppressor genes occurs as a consequence of hypermethylation within the promoter regions, and numerous studies have demonstrated a broad range of genes silenced by DNA methylation in different cancer types [72]. Epigenetic silencing is one of the mechanisms responsible for PROX1 inactivation in tumors. For example, hypermethylation of CpG islands was identified as a mechanism for PROX1 inactivation in breast, biliary system, and squamous cell carcinomas [11,13,14]. 

Our internal analysis using an online tool with clinical data—UCSC Xena database—has detected a heatmap showing relative methylation levels for the most variably methylated *PROX1* promoter regions in THC tissues (Figure 4A). Hierarchical clustering (from top to down) provides different THC tissues, where blue indicates a low level or no methylation for the reported locus of PROX1, and red shows the high methylation. Furthermore, we provide our experimental data, where two cell lines, normal thyroid cells (Nthy ori-3-1) and PTC-derived cell line (TPC-1), showed a significant increase in PROX1 transcript after 24 h of treatment with a demethylating agent (5-aza-2’-deoxycytidine; 5 μM) (Figure 4B).

Taken together, these data suggest that PROX1 expression could be regulated by DNA methylation status in thyroid cancer tissues and thyroid normal and cancer cells.

### 5.2. Non-Coding RNAs

Non-protein-coding RNAs (ncRNAs) have been associated with transcription/translation regulation and include microRNA (miRNA; approximate length 21–23 nucleotides) and non-protein-coding transcript (lncRNA; ≥ 200 nucleotides).

Several microRNAs have been shown to play critical roles in postnatal and pathologic angiogenesis [73] and pose attractive targets for the generation of novel therapeutic agents to treat vascular diseases and cancer [74]. 

It was demonstrated that the microRNA miR-181a is expressed in LEC cells and binds to the *PROX1* 3′-UTR, resulting in rapid and efficient transcript degradation and translational inhibition, which may have important implications for the control of PROX1 expression [68]. 

The miR-31 targets the 3’ UTR of *PROX1* to suppress its expression in human LEC cells, and conversely, miR-31 overexpression led to defective lymphangiogenesis in Xenopus and Zebrafish embryos [75]. Next, using LEC cells and alkali burn corneal injury model, it was shown that miR-466 directly targets the 3’ UTR of *PROX1*, and similarly to other miRNAs, suppresses PROX1 expression resulting in inhibition of lymphangiogenesis [76]. 

According to the miRDB database (http://www.mirdb.org/index.html), for example, miR-10527-5p, miR-6867-5p, miR-4262, miR-4668-5p, and miR-3148 may still regulate *PROX1* mRNA and thus can be the future research aim, especially in THC cases, due to the lack of published data.

The expression of PROX1 can be regulated by lncRNA (*PROX1-AS1*), which was also shown to be involved in THC biology [77]. According to research by Shen et al., 2018, *PROX1-AS1* is expressed in PTC-derived cell lines and regulates their malignant behavior [77]. In detail, knockdown of *PROX1-AS1* significantly inhibited proliferation, colony formation, migration, and invasion of PTC cells. Moreover, detected changes were associated with the mesenchymal-to-epithelial transition, where *PROX1-AS1* downregulation lowered the expression of N-cadherin and Vimentin, while E-cadherin was enhanced [77]. 

Using the GEPIA database, we detected ~4× higher *PROX1-AS1* expression in cancer samples (THC, *n* = 512) compared to healthy tissues (*n* = 337) (Figure 5A). Similarly to *PROX1*, the lower *PROX1-AS1* expression is associated with a higher tendency to the longer survival time of THC patients (Figure 5B), and a higher expression level of *PROX1-AS1* is observed in lower stages of the tumor (Figure 5C; [22]). The significant positive *PROX1*: *PROX-AS1* correlation (0.75; *p* = 0) was observed for THC samples (Figure 5D) and healthy tissues (0.5; *p* = 0, Figure 5E).

## 6. The Role of the PROX1 in the Regulation of Cancer Biological Processes, Including Thyroid Cancer

PROX1 is involved in the stimulation of multiple intracellular signaling pathways regulating apoptosis, proliferation, lymph-/angiogenesis, and EMT of cancer cells. Thus, PROX1 in neuroblastomas and pancreas occurs in a subset of well-differentiated, high-grade tumors [78,79]. PROX1 overexpression enhanced the proliferation of glioblastoma cells and promoted the growth of glioblastoma xenograft tumors, and this invasiveness potential was regulated via activation of the NF-κB signaling pathway [79]. In esophageal squamous cell carcinoma cell lines, the PROX1 protein was expressed at a lower level compared with the healthy exocrine pancreas [10], and low expression was associated with poor patient survival. Contrary, in colorectal cancer, depletion of PROX1 in human hepatocellular carcinoma cell lines caused a significant increase in cell proliferation [80]. 

In colorectal cancer, PROX1 has been identified as a downstream target of the TCF/beta-catenin signaling pathway [30], and its lower expression in colon cancer cells was connected with estrogen receptor beta signaling [81]. While PROX1 does not appear to be responsible for the initiation of colon tumor cell growth, it promotes progression from a benign to a malignant phenotype [30,82]. Analysis of the PROX1 regulatory pathways showed that this phenotypical switch is most likely induced through alterations in cell polarity, extracellular matrix interactions, cell adhesion, and is associated with dysplasia and frequent mitotic figures [30].

A study of Kaposiform hemangioendothelioma revealed that overexpression of PROX1 facilitates a more aggressive behavior through induction of genes involved in cell adhesion, proteolysis, and migration, thereby enhancing cell invasion and migration into the surrounding tissue [17].

Overall, the examples mentioned above provide indirect evidence that PROX1 may regulate tumor progression by influencing cancer cell migration and invasion.

In the context of THC, PROX1 can be an essential regulator of secretory granules (SGs) formation in MTC. Its presence was observed in SGs in immunohistochemistry staining, and *PROX1* gene depletion resulted in the reduced SG numbers and decreased expression of SG-related genes (Chromogranin A, Chromogranin B, Secretogranin II, Secretogranin III, Synaptophysin, and Carboxypeptidase E). Conversely, the introduction of a PROX1 transgene into a PTC and ATC cells induced the expression of SG-related transcripts [83].

The downregulation of PROX1 in PTC-derived cells was a consequence of aberrantly activated Notch signaling. Moreover, in PTC cells after transgenic PROX1 reexpression enhanced Wnt/β-catenin signaling was observed, coupled with and regulation of thyroid cancer 1 (TC-1) protein, Serpina 1, and Fatty acid-binding protein 4 (FABP4), that are known to be associated with PTC. Additionally, notch-induced PROX1 inactivation significantly promoted the malignant phenotype of thyroid cancer cells [15]. On the other hand, we observed that in FTC- and SCT-derived cells, PROX1 acts as an oncoprotein and supports malignant traits, including migration and invasion potential, anchorage-independent growth, that were accompanied by changes in focal adhesion force [20,21]. Furthermore, PROX1 knockdown increased the angiogenic potential of FTC- and SCT-derived cells by modulating the expression of genes involved in the angiogenic signaling pathway and was regulated in the opposite direction than pro-angiogenic factor FGF2 [22,84]. We can hypothesize that the discrepancy between the regulation of PTC and FTC and SCT by PROX1 may result from differences in the origin of cancer cells, mutations, as well as signaling pathways involved.

In our previous research, we analyzed the pattern of global gene expression in CGTH-W-1 cells lacking the PROX1 after transfection with RNA interference targeting of *PROX1*. We found that transcripts of many genes involved in migration, focal adhesion, invasion, cytoskeleton reorganization, and angiogenesis were regulated by PROX1 knockdown compared to control treated with negative siRNA. Studies examining biological processes have confirmed the involvement of selected factors as regulators of biological processes connected with cell–cell adhesion, cell migration, invasive behavior, and tube formation. Further, all molecular changes were rigorously confirmed in biological testing where cells after PROX1 knockdown revealed changed actin organization, showed the lower motility, increased invasive potential, and changed the tubularization of human umbilical endothelial cells cultivated in medium conditioned using CGTH-W-1 cells transfected with si*PROX1*. 

Here, we provide a review of other biological processes (BPs) significantly changed by downregulated (Table 1) and upregulated (Table 2) genes in PROX1-silenced CGTH-W-1 cells, and 20 transcripts showing the strongest differential expression (Table 3). The altered BPs are strongly connected with the aspect of transcriptional and gene regulation, translation, and neuro-/morphogenesis control. Table 3 presents selected genes and their role in the physiological and tumorigenic context, including function in thyroid cancer development, if data are available. Among the listed genes: *MMP1, VPS33A, CARNMT1, RASSF2, SOX2, MXRA5, SEPT3, LPAR1, FAM129A, FTH1, TUBA1* have already been connected with THC development, but the role of others remains unknown for THC biology. Therefore, the provided data can be a base for the perspective research focused on thyroid cancer and the PROX1.

## 7. Relation of PROX1 with Other Lymphatic Factors

Although the importance of PROX1 in lymphangiogenesis is widely described, little is known about the mechanisms by which PROX1 expression is controlled in other cell types and how PROX1 affects other genes/proteins. 

The road of lymphangiogenesis depends on VEGF-C and -D signaling pathways through VEGFR-2 and VEGFR-3, where especially VEGF-C and –D bind with VEGFR-3 and activate PROX1. In the cardinal vein, the PROX1-positive precursor cells differentiate into LEC cells. During asymmetrical division, a one daughter cell becomes lymphatic and progressively upregulates PROX1, and the other one downregulates PROX1 and stays in the vein [124]. In this process, VEGF-C controls the bipotential precursor division and mediates activation of VEGF-3, which next regulates PROX1 by establishing a feedback loop. Both VEGF-C and VEGFR-3 are required to PROX1-mediated cell fate reprogramming and to maintain the identity of LEC progenitors [125]. Additionally, the PROX1 activation of VEGFR-3 can be regulated by small ubiquitin-like modifier 1 (SUMO-1; sumoylation), which can reduce PROX1 transcriptional activity and consequently stops the lymphatic differentiation [126]. 

The PROX1/VEGF-C/VEGFR3 positive correlation was also detected in the analysis of human cervical neoplasia [127]. Furthermore, in vitro analysis demonstrated that PROX1 regulates cell growth, proliferation, invasion, and lymphangiogenesis by enhanced VEGFC expression in oral squamous cell carcinoma [128]. In thyroid cancer cell lines (FTC-133 and CGTH-W-1), PROX1 and VEGFC are expressed, and in CGTH-W-1 cells, VEGFC was oppositely regulated by PROX-1 knockdown, which enhanced the VEGFC expression level and its secretion [22]. 

On the other hand, PROX1 can be directly activated by the transcription factors SOX18 [129] and COUP-TFII [130,131], binding to the PROX1 promoter. Overexpression of SOX18 in blood endothelial cells induces them to express PROX1 and other lymphatic endothelial markers, while Sox18-null embryos show a complete blockade of LEC cells differentiation from the cardinal vein [129] and loss of PROX1 expression [132]. Furthermore, defects in the transcription factor SOX18 cause lymphatic dysfunction in the human syndrome hypotrichosis–lymphoedema–telangiectasia [133].

Throughout LECs development, COUP-TFII physically interacts with PROX1 to form a stable complex. As a result, COUP-TFII is a partner for PROX1 to control several genes, including VEGFR-3, FGFR-3, and neuropilin-1, required to LEC phenotype [134].

Similarly to PROX1, both SOX18 and COUP-TFII are associated with regulation in malignancy of various cancer types, such as gastric, breast, and lung cancers, and they are connected with vascularization of tumors [135,136,137,138].

PROX1 enables the reprogramming of vascular endothelial cells to become PDPN-expressing lymphatic endothelial cells [6,139,140]. PDPN—a mucin-type transmembrane protein—is a unique transmembrane glycoprotein receptor (38 to 50 kDa) with a heavily O-glycosylated amino-terminal extracellular domain. PDPN does not show enzymatic activity; to accomplish its biological functions, PDPN interacts with other proteins located in the same cell or neighbor cells [141]. Consequently, the binding of PDPN to its ligands leads to the modulation of signaling pathways that regulate proliferation, contractility, migration, EMT, and remodeling of the extracellular matrix [142]. In the PTC-derived cells, PDPN silencing reduces migration, invasion, and adhesion of tested cells through regulating the expression of the ezrin, radixin, and moesin proteins, MMP9 and MMP2 proteins [32,33].

Using chromatin immunoprecipitation assay, performed on LEC cells, PROX1 binding to the 5′ regulatory sequence of *PDPN* regulating *PDPN* mRNA expression level was detected (Figure 6A) [105]. In the context of THC, PROX1 and PDPN expression is adjusted in opposite directions. PROX1 more highly expressed in FTC and SCT cells, whereas PDPN shows the upregulation in PTC cells [32]. Furthermore, in THC tissues, the positive co-expression of PROX1 and PDPN was on a slight level (= 0.0052) and statistically insignificant (Figure 6B). 

All the above observations can suggest that PROX1 relation with other vascular factors should be separately investigated in THC cases.

## 8. Conclusions and Perspectives

PROX1 shows a vital role in tumor progression and metastatic tumor growth through the impact on the aggressiveness of various cancer types, including thyroid cancer.

However, more profound knowledge regarding the molecular mechanisms, pathways, and targets of PROX1 in the various stages of thyroid tumor development remains to be obtained. Molecular regulation of PROX1 was deeply investigated in LEC cells, but it is unknown how PROX1 regulates and is regulated in other types of cells, especially under tumorigenic events.

Our work highlights the utility of PROX1 as a potential prognostic marker and adds biological insight to its role in different thyroid cancers, where it controls a gene expression profile involved in migration, invasion, adhesion, and vascularization. Still, further experiments are required to understand how the PROX1 epigenetic regulation and relation with other vascular molecules translate into a tumor setting and development. Considering the aggressive nature of THC and the minimal treatment window, there is an imperative need for novel molecular-based treatment strategies in which PROX1 can be an essential factor.

## Figures and Tables

**Figure 1 ijms-21-03220-f001:**
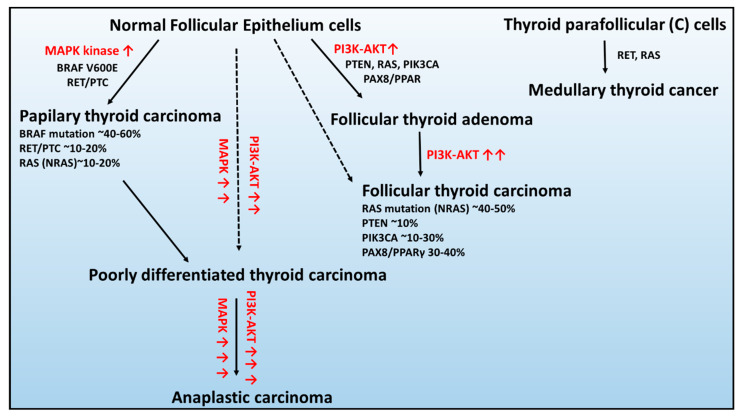
Schematic of the genetic abnormalities associated with the development and progression of thyroid cancers developed from healthy epithelium cells: (**1**) follicular thyroid adenoma, (**2**) papillary thyroid carcinoma, (**3**) follicular thyroid carcinoma, and (**4**) anaplastic thyroid carcinoma and parafollicular thyroid cells: medullary thyroid carcinoma. Figure adapted from [45]. Red arrow signifies overactivation of signaling pathway.

**Figure 2 ijms-21-03220-f002:**
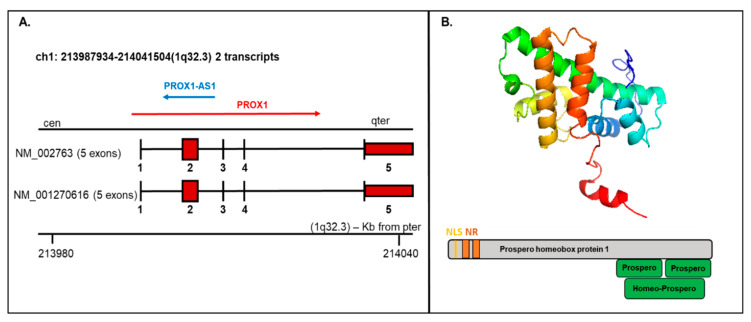
Prospero homeobox 1 (PROX1) mRNA/protein structure. (**A**) The *PROX1* transcript presents two variants (1-NM_002763 and 2-NM_001270616; the first variant represents the longer transcript; both options 1 and 2 encode the same protein); mRNA includes five exons and four introns. The structure is available on http://atlasgeneticsoncology.org/Genes/GC_PROX1.html (**B**) Top—Protein structure-based on the PDB model (ID: 2LMD; graphical visualization was made in PyMOL), bottom—the schematic protein structure with the nuclear localization signal (NLS) and nuclear receptor (NR) boxes at the N-terminus and prospero and homeobox domains on C-terminus.

**Figure 3 ijms-21-03220-f003:**
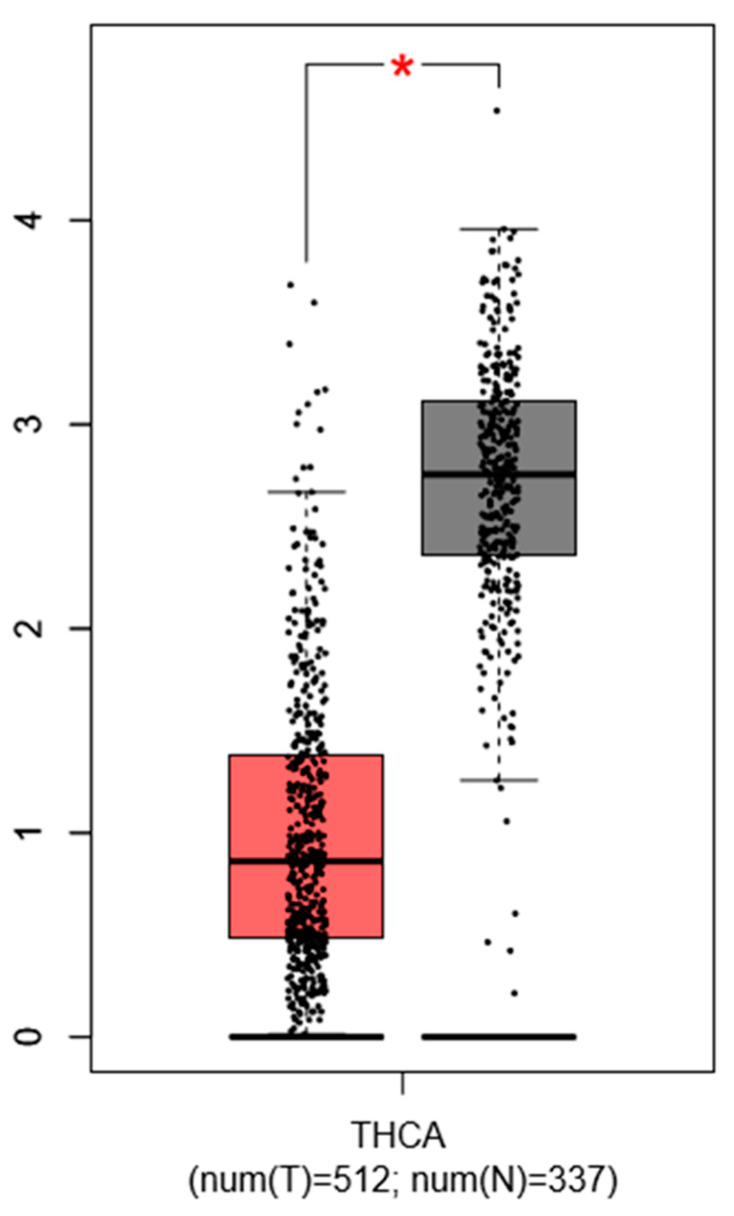
Expression pattern of *PROX1* in thyroid cancer (THC) tissues. The red bar represents the *PROX1* expression in papillary thyroid tumors (T; the number of cases = 512) and grey bar shows the *PROX1* level in healthy tissues (N; the number of cases = 337), * *p* ≤ 0.05. The data are available on the GEPIA database.

**Figure 4 ijms-21-03220-f004:**
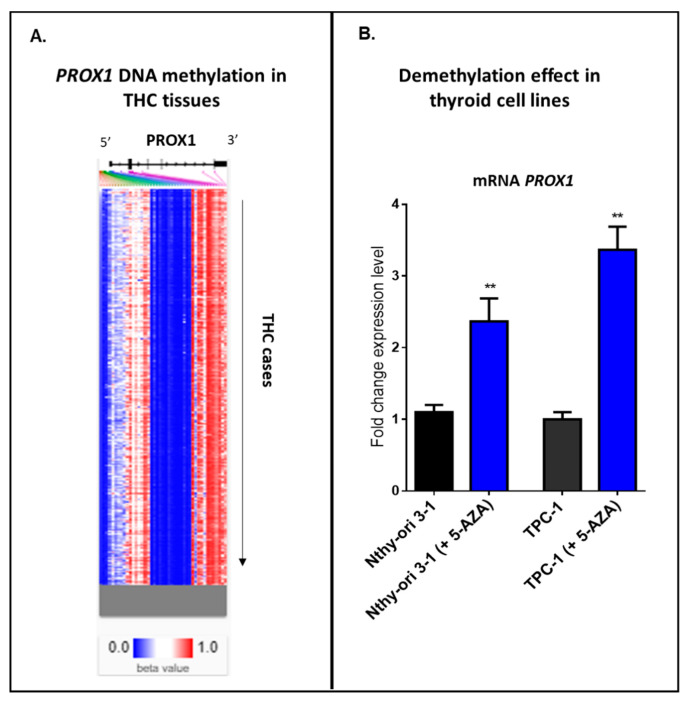
Methylation in THC cases. **A.** Hierarchical clustering of THC tissues; blue indicates a low level or no methylation for PROX1; red indicates a high methylation level. The methylation analysis was studied using https://xenabrowser.net/heatmap/. **B.** Treatment of the immortalized normal thyroid cells (Nthy ori-3-1) and PTC-derived cell line (TPC-1) with 5 µM 5-aza-2’-deoxycytidine for 24 h, ** *p* ≤ 0.01.

**Figure 5 ijms-21-03220-f005:**
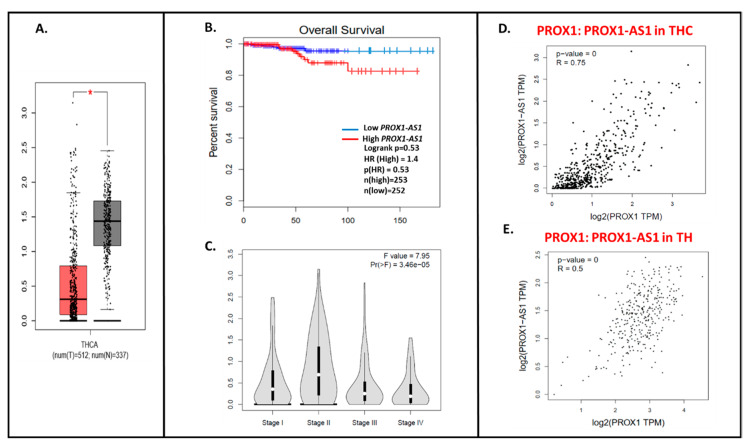
The *PROX1-AS1* expression pattern in thyroid cancer and correlation with *PROX1*. (**A**) Expression of *PROX1-AS1* in thyroid cancer specimens (pictured on the red bar (T); the number of cases = 512) compared to non-cancerogenic thyroid tissues (pictured on the grey bar (N); the number of cases = 337), * *p* ≤ 0.05. (**B**) *PROX1-AS1* relation to survival of patients with THC. (**C**) *PROX1-AS1* expression compared to tumor stage I, II, III, and IV. (**D**,**E**) *PROX1-AS1*: *PROX1* correlation in THC and healthy thyroid (TH) specimens, respectively.

**Figure 6 ijms-21-03220-f006:**
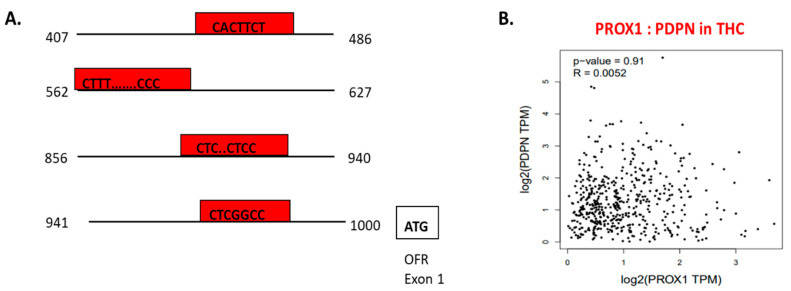
The transcriptional function of PROX1 and interaction with other factors. **A.** PROX1 is a transcription factor of PDPN [143]; *PROX1* binds to the 5’ regulatory sequence of *PDPN* and regulates PDPN expression. The graph presents a 5′ regulatory DNA region of the human *PDPN* gene and putative cis-elements (regions of non-coding DNA that binding to transcription factors; in red boxes) for PROX1. **B.**
*PROX1*–*PDPN* relation in papillary thyroid cancer tissues. Data are available on https://www.grnpedia.org/trrust.

**Table 1 ijms-21-03220-t001:** Gene Ontology (biological process) analysis of up-regulated genes after Prospero homeobox 1 (PROX1) silencing in CGTH-W-1 cells.

GO ID	GO Name	*p*-Value	Count
GO:0006413	translational initiation	1.60 × 10^−19^	35
GO:0006415	translational termination	2.06 × 10^−17^	31
GO:0006414	translational elongation	2.30 × 10^−17^	35
GO:0032984	macromolecular complex disassembly	2.43 × 10^−17^	61
GO:0006614	signal-recognition particle -dependent cotranslational protein targeting to membrane	7.11 × 10^−17^	35
GO:0072599	establishment of protein localization to the endoplasmic reticulum	9.42 × 10^−16^	35
GO:0044419	interspecies interaction between organisms	4.95 × 10^−15^	91
GO:0044033	multi-organism metabolic process	6.53 × 10^−15^	47
GO:0000184	nuclear-transcribed mRNA catabolic process, nonsense-mediated decay	9.84 × 10^−14^	33
GO:0006612	protein targeting to membrane	4.70 × 10^−13^	36
GO:0006402	mRNA catabolic process	2.46 × 10^−11^	41
GO:0034660	non-coding RNA metabolic process	2.78 × 10^−11^	39
GO:0071822	protein complex subunit organization	4.71 × 10^−11^	59
GO:0044238	primary metabolic process	5.31 × 10^−9^	224

**Table 2 ijms-21-03220-t002:** Gene Ontology (biological process) analysis of down-regulated genes after PROX1 silencing in CGTH-W-1 cells.

GO ID	GO name	*p*-Value	Count
GO:0072359	circulatory system development	8.73 × 10^−8^	56
GO:0007275	multicellular organismal development	1.10 × 10^−7^	86
GO:0032879	regulation of localization	5.58 × 10^−7^	31
GO:0022008	neurogenesis	1.32 × 10^−6^	44
GO:0061564	axon development	3.51 × 10^−6^	37
GO:0007173	epidermal growth factor receptor signaling pathway	3.43 × 10^−5^	17
GO:0048812	neuron projection morphogenesis	4.84 × 10^−5^	34
GO:0051272	positive regulation of cellular component movement	5.25 × 10^−5^	24
GO:0048667	cell morphogenesis involved in neuron differentiation	5.53 × 10^−5^	33
GO:0030910	olfactory placode formation	0.000117108	4
GO:0071698	olfactory placode development	0.000117108	4
GO:0050896	response to stimulus	0.000131862	86
GO:0021707	cerebellar granule cell differentiation	0.000158245	3
GO:0009605	response to external stimulus	0.000216347	53

**Table 3 ijms-21-03220-t003:** List of 20 genes up- and downregulated in CGTH-W-1 cells upon PROX1-knockdown.

GO ID	Fold Change	*p*-Value	Function
***MMP1*** (Matrix metallopeptidase 1)	4.025	4.30 × 10^−22^	An enzyme responsible for the extracellular matrix remodeling [85] and can play a significant role in the invasion and recurrence of thyroid cancer (THC). Contrary to healthy thyroid cells, MMP-1 is secreted by thyroid cancer cells in vitro, and immunofluorescence staining showed the expression of MMP-1 protein in adenomas, papillary thyroid cancer (PTC) and follicular thyroid cancer (FTC) specimens [86,87]. Interestingly, an examination of the *MMP1* mRNA in PTC cases revealed that the transcript was expressed in the fibrous capsules of PTC, but not in the PTC cells [88].
***ZBED2*** (Zinc finger BED-type containing 2)	2.911	2.84 × 10^−9^	The transcription factor regulating cellular signals in a cell-type specific manner [89]. *ZBED2* was detected in PTC tissues as a commonly downregulated gene [90]; nonetheless, its role is unknown in THC development.
***VPS33A*** (VPS33A core subunit of CORVET And HOPS complexes)	2.360	1.42 × 10^−6^	Molecule involved in the endocytic membrane transport and autophagic pathways [91]. According to mRNA expression, *VPS33A* can be considered as a favorable prognostic marker in THC [92].
***NDUFA2*** (NADH: ubiquinone oxidoreductase subunit A2)	2.337	1.11 × 10^−7^	A unit of the hydrophobic protein fraction of the NADH: ubiquinone oxidoreductase (complex 1), which catalyzes the first step in the mitochondrial respiratory chain; translocate electron across the inner mitochondrial membrane [93]. NDUFA2 can be a useful marker in discriminating healthy and highly invasive breast carcinoma [94].
***TSEN54*** (TRNA splicing endonuclease subunit 54)	2.315	2.38 × 10^−5^	A part of the splicing endonuclease complex involved in tRNA splicing and mRNA 3′ end formation [95]. TSEN54 is scored as one of the important SEN subunits in multiple human haploid cancer cell lines [96].
***CACNA2D3*** (Calcium voltage-gated channel auxiliary subunit alpha2delta 3)	2.285	7.69 × 10^−8^	Member of the voltage-dependent calcium channel complex. CACNA2D3 can act as a cancer suppressor, e.g., in glioblastoma [97] and endometrial cancer [98].
***HIST1H4A*** (H4 clustered histone 1)	2.232	3.30 × 10^−9^	A protein is belonging to the linker histone family that is engaged in chromatin organization. Its alteration in expression level can be connected with cancer aggressiveness [99].
***CARNMT1*** (Carnosine N-methyltransferase 1)	2.218	0.000425	A cytosolic enzyme catalyzing the N-methylation of nicotinamide to form 1-methylnicotinamid, which plays an essential role in controlling the intracellular concentration of nicotinamide. CARNMT1 is upregulated in various cancers [100]. Its activities were detected in 8 of 10 of PTC-derived cell lines and 3 of 6 of the FTC-derived cell lines. Immunohistochemical labeling showed abundant cytoplasmic reactions in the sections of PTC, and scanty reaction in the control of thyroid tissues [101]. Furthermore, healthy thyroid tissues, primary thyroid cultures, anaplastic thyroid cancer (ATC) cells, and medullary thyroid cancer cells showed no or low enzyme activity.
***SLCO4A1*** (Solute carrier family 4 member 1 (Diego blood group))	2.179	0.000354	Protein involved in the transport of various compounds, such as sugars, bile salts, organic acids, metal ions, amine compounds, and estrogen. SLCO4A1 is highly expressed in several tumors, e.g., breast, colorectal, and lung cancers [102].
***RBM8A*** (RNA binding motif protein 8A)	2.170	7.44 × 10^−12^	An RNA binding protein is a component of the exon junction complex. Abnormal RBM8A expression is associated with carcinogenesis [103].
***RASSF2*** (RAS association domain family member 2)	0.403	2.04 × 10^−7^	The RAS association domain family encodes the class of tumor suppressors. Under the tumorigenic transformation, several RAS members are frequently silenced in human cancer. In this context, *RASSF2* was methylated in 88% of thyroid cancer tissues, varied THC-derived cell lines, and 63% of primary thyroid carcinomas [104].
***SOX2*** (SRY-box transcription factor 2)	0.411	6.61 × 10^−6^	SOX2 is essential for embryonic development and maintaining the stemness of embryonic cells. On the other hand, the deviation of SOX2 expression positively correlates with the enhancement of cancer cell traits, such as proliferation, migration, invasion, and drug resistance [105]. In the context of thyroid cancer, the higher expression of SOX2 can associate with the transformation from PTC to ATC accompanied by tumor protein p53 (*TP53*) mutation [106].
***MXRA5*** (Matrix remodeling associated 5)	0.416	3.02 × 10^−6^	The role of the MXRA5 protein is not clear; however, its anti-inflammatory and anti-fibrotic functions were shown in tubular cells of the human renal biopsies [107]. The *MXRA5* mRNA mutation was detected in some cases of follicular variant of papillary thyroid cancer [108].
***SEPT3*** (Septin 3)	0.423	2.74 × 10^−12^	A cytoskeletal GTPase involved in many cellular processes, including exocytosis, apoptosis, carcinogenesis, and neurodegeneration. It can suppress the growth of some tumors, including glioma and PTC [109].
***LPAR1*** (Lysophosphatidic acid receptor 1)	0.423	2.26 × 10^−14^	The receptor of the G protein-coupled that mediates diverse biologic functions, such as proliferation and survival of cells, cytoskeleton reorganization, factor secretion, and tumor cell invasion. LPAR1 expression level is induced in THC-derived cell lines (BcPAP, SW173, CAL62), compared to normal thyroid epithelium cells (Nthy-ori 3-1). In the THC cells, LPAR1 mediates invasion through the RHOA and ERK signaling pathway, which is amplified by heterodimerization with a member of the adhesion G protein-coupled receptor family—CD97 [110].
***FAM129A*** (Niban apoptosis regulator 1)	0.424	3.79 × 10^−13^	FAM129A protein is activated in the stress conditions (e.g., ER-stress and genotoxic stress) in cells and protects them from apoptosis and death [111]. FAM129A regulates autophagy in a cell/context-dependent manner, increasing or decreasing autophagocytosis activity in healthy and tumor thyroid cells, respectively [112]. Immunohistochemistry staining of FAM129A does not reliably distinguish follicular thyroid carcinoma from follicular thyroid adenoma [113]. However, it was described that it can be a useful marker to distinguish benign from malignant thyroid nodules in preoperative diagnostic exams [114].
***FTH1*** (Ferritin heavy chain 1)	0.434	4.15 × 10^−23^	A vital ferritin subunit is maintaining iron balance, which can act as a suppressor in breast colorectal and ovarian cancer [115,116]. On the other hand, FTH1 transcript overexpression was detected in a two-step differential expression analysis of six Hürthle cell follicular thyroid adenoma and their paired normal tissues [117].
***TUBA1A*** (Tubulin alpha 1a)	0.446	2.05 × 10^−21^	Represent the major components of cytoskeleton microtubules. The post-translational modifications of tubulin have been reported for a range of cancers and correlated with poor prognosis and chemotherapy resistance in cancer treatment [118]. Greatly increased expression of *TUBA1A* mRNA was detected in PTC and anaplastic carcinomas [119]. Furthermore, TUBA1A protein was defined as one of the predictive markers in differentiating between FTC and adenoma or between FTC and PTC [120].
***PNMA2*** (PNMA family member 2)	0.448	4.67 × 10^−6^	PNMA2 is mainly expressed in the healthy human brain, but its presence was also found in other human tumors. The physiological function of PNMA2 is still unclear; however, the investigation using cell line derived from breast cancer (MCF-7) point out that PNMA1 promotes apoptosis and chemo-sensitization [121].
***FBXO32*** (F-box protein 32)	0.449	3.57 × 10^−5^	F-box protein an E3 ubiquitin ligase that is involved in phosphorylation-dependent ubiquitination and plays remarkable roles in tumorigenesis and muscle atrophy [122]. FBXO32 can act as a suppressor in breast and ovarian cancer cells by regulating the malignant behavior of tested cells [122,123].

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
