# Peer review of "The Impact of Transcription Factor Prospero Homeobox 1 on the Regulation of Thyroid Cancer Malignancy"

_ijms, 2020, doi:10.3390/ijms21093220_

Round 1
Reviewer 1 Report
Manuscript ijms-781805
Suggestions/Comments:
The structure and flow of the text can be further reorganized. The authors shall link ideas into a logical flow. Although one or two-sentence paragraph is alright but will not be a well-organized body. However, many of these were read. Furthermore, the smooth flow of sentences can be further improved.
Author Response
Thank you for your beneficial suggestions. We have adequately addressed your concerns below and have thoroughly revised the manuscript based on your recommendations.
The structure and flow of the text can be further reorganized. The authors shall link ideas into a logical flow. Although one or two-sentence paragraph is alright but will not be a well-organized body. However, many of these were read. Furthermore, the smooth flow of sentences can be further improved.
Author response: We did our best to improve the flow of the text. We rebuilt the paragraphs and e.g., lines: 47, 94, 105,108, 113, 121, 124, 128, 301, 305, 312 were combined with the rest of the text, to avoid 2-3 sentences in separate paragraphs.
Reviewer 2 Report
The work submitted by authors Rudzinska and Czarnocka, is detailed and is a well-organized review on the structure, function, epigenetic regulation, expression, and role of the transcription factor PROX1 in thyroid cancer. They have reviewed the PROX1 mRNA/protein structures and the biological processes in detail and its relation with other lymphatic factors. The work has to potential to contribute to further studies looking into the role of PROX1 as a biological marker in thyroid cancer.
I have few minor comments for the authors:
- TERT promoter mutations have been widely studied in aggressive thyroid cancer (Liu et al. 2013 - https://www.ncbi.nlm.nih.gov/pubmed/23766237) and has been established to confer poor outcomes in PTC (Shen et al., 2017 https://www.ncbi.nlm.nih.gov/pubmed/27875244). It will be valuable to highlight the role of TERT promoter mutations in aggressive thyroid cancer in section 2 under thyroid cancer classification.
- It is worthwhile to note that in an interesting review by Lee et al. , 2019 - https://www.ncbi.nlm.nih.gov/pmc/articles/PMC6759718/, they had unexpectedly found TERT and PROX1 mRNA expression levels were correlated in several cancer types including melanoma, esophageal and head and neck and lung cancer, with PROX1 downregulation indicating a poorer prognosis in melanoma. The authors concluded that PROX1 perhaps regulates TERT in an activity dependent manner with other genetic changes.
- Line 90 spelling error, FTC ceases to FTC cases
- Figure 1 Spelling error, within the figure under Follicular thyroid carcinoma: RAX8/PPARg to PAX8/PPARg
Author Response
Dear Reviewers
Thank you for your beneficial suggestions. We have adequately addressed your concerns below and have thoroughly revised the manuscript based on your recommendations.
I have few minor comments for the authors:
- TERT promoter mutations have been widely studied in aggressive thyroid cancer (Liu et al. 2013 - https://www.ncbi.nlm.nih.gov/pubmed/23766237) and has been established to confer poor outcomes in PTC (Shen et al., 2017 https://www.ncbi.nlm.nih.gov/pubmed/27875244). It will be valuable to highlight the role of TERT promoter mutations in aggressive thyroid cancer in section 2 under thyroid cancer classification.
It is worthwhile to note that in an interesting review by Lee et al. , 2019 - https://www.ncbi.nlm.nih.gov/pmc/articles/PMC6759718/, they had unexpectedly found TERT and PROX1 mRNA expression levels were correlated in several cancer types including melanoma, esophageal and head and neck and lung cancer, with PROX1 downregulation indicating a poorer prognosis in melanoma. The authors concluded that PROX1 perhaps regulates TERT in an activity dependent manner with other genetic changes.
Author response: Thank you for this significant comment. We have included pointed references and information about TERT: PROX1 relation (lines 138-152).
- Line 90 spelling error, FTC ceases to FTC cases
Author response: It has already been corrected.
- Figure 1 Spelling error, within the figure under Follicular thyroid carcinoma: RAX8/PPARg to PAX8/PPARg
Author response: It has already been fixed.